# Fetal Biometric Assessment and Infant Developmental Prognosis of the Tadalafil Treatment for Fetal Growth Restriction

**DOI:** 10.3390/medicina59050900

**Published:** 2023-05-08

**Authors:** Makoto Tsuji, Shintaro Maki, Naosuke Enomoto, Kota Okamoto, Asa Kitamura, Shoichi Magawa, Sho Takakura, Masafumi Nii, Kayo Tanaka, Noriko Yodoya, Hiroaki Tanaka, Hirofumi Sawada, Eiji Kondo, Masahiro Hirayama, Tomoaki Ikeda

**Affiliations:** 1Department of Obstetrics and Gynecology, Mie University Graduate School of Medicine, Edobashi, Tsu 2-174, Mie, Japan; 2Department of Pediatrics, Mie University Graduate School of Medicine, Edobashi, Tsu 2-174, Mie, Japan

**Keywords:** tadalafil, fetal growth restriction, fetal biometric assessment, fetal head circumference, brain-sparing effect

## Abstract

*Background and Objectives:* Tadalafil is expected to treat fetal growth restriction (FGR), a risk factor for stillbirth and neonatal morbidity. This study aimed to evaluate the fetal biometric growth pattern of fetuses with FGR treated with tadalafil by ultrasonographic assessment. Materials and *Methods*: This was a retrospective study. Fifty fetuses diagnosed with FGR and treated by maternal administration of tadalafil and ten controls who received conventional treatment at Mie University Hospital from 2015 to 2019 were assessed. Fetal biparietal diameter (BPD), head circumference (HC), abdominal circumference (AC), femur length (FL), and estimated fetal weight (EFW) at the start of treatment and at two weeks and four weeks of treatment were mainly assessed by ultrasound examination. The Wilcoxon signed-rank test was used to assess the measures. The Kyoto Scale of Psychological Development (KSPD) was used to assess the developmental prognosis on tadalafil-treated children at 1.5 years of corrected age (CA) and 3 years old. *Results*: The median gestational age at the start of treatment was 30 and 31 weeks in the tadalafil and control groups, respectively, and the median gestational age at delivery was 37 weeks in both groups. The Z-score of HC was significantly increased at 4 weeks of treatment (*p* = 0.005), and the umbilical artery resistance index was significantly decreased (*p* = 0.049), while no significant difference was observed in the control group. The number of cases with an abnormal score of less than 70 on the KSPD test was 19% for P-M, 8% for C-A, 19% for L-S, and 11% for total area at 1.5 years CA. At 3 years old, the respective scores were 16%, 21%, 16%, and 16%. *Conclusions*: Tadalafil treatment for FGR may maintain fetal HC growth and infants’ neuro-developmental prognosis.

## 1. Introduction

Fetal growth restriction (FGR) is a serious condition that is associated with various adverse outcomes and increased perinatal mortality [1,2,3,4,5,6,7]. Furthermore, chronic hypoxia and malnutrition may increase the risk of future developmental disorders such as autism, learning disorders, and lifestyle-related diseases such as hypertension and type 2 diabetes [8,9]. Since appropriate delivery according to fetal well-being is the only strategy for management of FGR, premature delivery of the fetus and severity of the FGR can be an issue.

The major cause of this serious condition is placental insufficiency, which leads to various perinatal complications such as hypertensive disorders of pregnancy (HDP), placental abruption, and preterm premature rupture of membranes [10]. Therefore, FGR is a serious disorder in obstetrics because there is no established, effective treatment. Various treatments for FGR have been considered, such as aspirin, heparin, and statins [11,12,13,14,15,16].

Phosphodiesterase 5 (PDE5) inhibitors block the phosphodiesterase enzyme, preventing the inactivation of the intracellular second messenger cyclic guanosine monophosphate within vascular smooth muscle cells, which potentiates the action of nitric oxide, leading to vasodilatation. These are candidates for new therapeutic agents to improve uteroplacental perfusion, and their efficacy and safety have been reported in several studies. The Sildenafil Therapy in Dismal Prognosis Early-onset Fetal Growth Restriction (STRIDER) consortium has been trying to provide evidence of the benefit of sildenafil citrate for FGR treatment [17,18,19,20,21,22,23,24]. The STRIDER UK study reported no benefits of sildenafil for pregnancy prolongation, survival, or short-term neonatal outcomes [18]. Similar STRIDER studies performed in New Zealand and Australia also reported that sildenafil did not affect the proportion of pregnancies with increase in fetal growth velocity [20]. The Dutch STRIDER study suspended the trial because its interim analysis showed an increased risk of neonatal persistent pulmonary hypertension [21] and concluded that sildenafil had no efficacy in reducing perinatal mortality or major neonatal morbidity [24].

Tadalafil is a PDE5 inhibitor that is effective and safe for FGR according to previous studies, including clinical trials and basic research [25,26,27,28,29,30]. Compared with sildenafil, tadalafil has longer a half-life and a more rapid onset of action [31] and less susceptibility to the intake of a high-fat meal [32,33]. In addition, it is thought that tadalafil and sildenafil have different mechanism for fetal–placental perfusion. Walton et al. reported that sildenafil citrate reversed pre-constricted placental–fetal arterial perfusion in an ex vivo human placental model, whereas tadalafil produced no response [34]. In our mice experiment, the width of the maternal blood sinuses in the placenta was significantly dilated by tadalafil administration; in contrast, there was no change in fetal capillary width with the administration [28]. This indicates that tadalafil does not cross the human placental barrier or is degraded by trophoblast cells and that the difference in the effects between tadalafil and sildenafil in the fetus via the placenta might result in differences in safety and efficacy.

For that reason, research for the efficacy of tadalafil on FGR has continued, as agents different from sildenafil yielded negative results. The results of a phase II randomized controlled trial (TADAFER II) showed a potential decrease in the mortality rate of FGR fetuses, neonates, and infants and the prolongation of pregnancy in early-onset FGR [25]. We started a randomized, placebo-controlled trial in September 2019 to establish further evidence for the efficacy of tadalafil because of the limitations of previous studies, including open-label design and trial suspension.

We hypothesized that tadalafil improves fetoplacental perfusion in FGR. In placental insufficiency, there is histological evidence of impaired trophoblastic invasion to spiral arteries in the placental bed, which leads to spiral arteries that remain narrowed [35]. Fetal hypoxemia induced by placental insufficiency produces a physiological response that increases blood flow to vital organs, including the brain, myocardium, and adrenal gland, as a “redistribution” [36,37,38,39,40]. The mechanism causes lowered abdominal circumference (AC) of fetuses due to reduced blood flow to the liver and glycogen stores. As gestational age progresses, and the mechanism cannot be kept up, it leads to stagnation of the head circumference (HC) growth, and this is thought to be associated with a worse neurological prognosis, including psychomotor and cognitive retardation in the child [41]. Therefore, if tadalafil causes an improvement in uteroplacental perfusion, changes in hemodynamics may affect the fetal growth pattern, including AC and HC.

This study evaluated the fetal growth pattern of tadalafil-treated fetuses with FGR. Additionally, we investigated the association between fetal development and postnatal neurological prognosis using neurodevelopmental tests.

## 2. Materials and Methods

### 2.1. Trial Design and Study Population

We conducted a retrospective study investigating the fetal growth pattern of mothers who received 10, 20, or 40 mg tadalafil for the treatment of FGR at Mie University Hospital between July 2015 and September 2019. This study included cases registered in our previous retrospective study [26] and in the phase I study [27]. Additional cases in which tadalafil had been administered upon the request of patients and that were not included in the two previous studies were also included in the present study.

FGR was defined as an estimated fetal body weight (EFBW) 1.5 standard deviations below the mean EFBW for the gestational age (GA; according to the Japanese standard curve). The GA of the fetus was calculated based on the last menstrual period or by the crown–rump length in the first trimester if a discrepancy exceeded 7 days. Data were collected for patients diagnosed with FGR by an obstetrician proficient in fetal ultrasound procedures in our facility. The patients were managed upon admission, and the data were collected. The following exclusion criteria were applied at the time that tadalafil was offered: (i) fetuses with suspected chromosomal disorders, multiple congenital anomalies, or viral infection; (ii) multifetal pregnancy; (iii) GA > 34 + 0 weeks; (iv) history of allergy to tadalafil or concurrent medications that interact with tadalafil; and (v) relative contraindications to tadalafil due to clinical conditions such as renal or liver disease, uncontrolled hypertension (blood pressure > 170/110 mmHg) or arrhythmia, coagulation defects, active gastric or intestinal ulcers, retinitis pigmentosa, and veno-occlusive disease.

The clinical study of tadalafil was originally inspired by a report [42] showing favorable fetal development in pregnancies with pulmonary hypertension with a high incidence of FGR. Insurance coverage in Japan for pulmonary hypertension covered tadalafil doses of 20 mg and 40 mg, which are known to be safe for pregnant women, and 10 mg, which is indicated for erectile dysfunction. Therefore, 10 mg, 20 mg, and 40 mg of tadalafil have been used in studies. Tadalafil was orally administered at 11:00 a.m. every day until delivery.

We also investigated pregnant Japanese women matched for maternal age, parity, and GA when diagnosed with FGR between July 2015 and September 2019 (control group) at Mie University Hospital. These women received conventional management of FGR according to the guidelines for obstetrical practice in Japan [43], which recommend that timing of termination should be determined by assessing fetal well-being evaluated by ultrasound and non-stress test.

### 2.2. Ultrasound Procedures

The examinations were performed between 9:00 a.m. and 5:00 p.m. because routine fetal biometry measurements were performed during the day shift. The patients underwent the test in the semi-Fowler’s position after adequate rest. EFBW, HC, biparietal diameter (BPD), AC, and femur length (FL) were included in the fetal ultrasonographic biometrical assessments. A transabdominal high-quality ultrasound system (Voluson E8; General Electric, Boston, MA, USA) was used for the examination.

EFBW was calculated according to the Japan Society of Obstetrics and Gynecology guidelines using a local chart [44]. The reference values for BPD, AC, and FL were based on the Japanese reference values created by Shinozuka et al. [44,45,46]. The HC value refers to the Japanese data by Kuno et al. [47]. BPD was measured from the proximal echo of the fetal skull to the proximal edge of the deep border (outer-inner) at the level where the cavum septum pellucidum breaks the midline echo approximately one-third of the way from the anterior border of the skull [45]. The HC was measured as an ellipse around the perimeter of the fetal skull at the same level [48]. The AC was measured by an ellipse tracing of the outer edge of the abdominal wall at right angles, where the umbilical vein was clearly visualized at a point one-third to one-quarter of its length from the anterior wall [45]. The FL was measured using the full femoral diaphysis and taken from one end of the diaphysis to the other, not including the distal femoral epiphysis [45,49].

Doppler assessments were also performed on the umbilical artery (UA) and middle cerebral artery (MCA) during the fetal biometric measurements. UA Doppler waveforms were measured in free loop of the umbilical cord. MCA Doppler wave-forms were obtained from the proximal portion of the vessel immediately near the circle of Willis, which showed the best reproducibility [50]. Resistance index (RI) was calculated for both vessels.

The examination was performed at start of treatment and 2 and 4 weeks later (±2 days), and the fetal growth patterns were evaluated. The ultrasonographic measurements at 2-week intervals were collected and analyzed to accurately assess interval growth [51,52].

### 2.3. Maternal Hemodynamics

Maternal systolic blood pressure (BP), diastolic BP, mean arterial blood pressure (MAP), and heart rate (HR) before the start of treatment and at 2 weeks and 4 weeks of treatment were assessed to evaluate the change in maternal hemodynamics due to maternal tadalafil administration. All measurements were conducted with participants in the sitting position after at least 5 min of rest using automated, non-invasive blood pressure monitoring devices (HBP-1600^®^, OMRON, Kyoto, Japan). Data after waking up and before taking oral medication were collected.

### 2.4. The Kyoto Scale of Psychological Development (KSPD) Test

We conducted a developmental evaluation with the KSPD test (published in 2002) at the corrected age (CA) of 1.5 years and 3 years old in cases treated with tadalafil [53]. The KSPD test is an individualized test performed in person by experienced psychologists to assess a child’s development in the following three areas: postural–motor functions (P–M; fine and gross motor functions); cognitive–adaptive functions (C–A; non-verbal reasoning or visuospatial perceptions assessed using materials); and language–social functions (L–S; interpersonal relationships, socializations, and verbal abilities). In each of the three areas, a sum score is converted to an overall developmental age (DA). The DAs for the three specific areas and the overall DA are divided by the child’s chronological age and multiplied by 100 to yield four developmental quotients (DQ). Infants are assessed as normal (DQ ≥ 85), borderline (70 < DQ < 85), or abnormal (DQ ≤ 70), when compared with ordinary children of the same age. In this study, experienced physicians and a speech pathologist performed the behavioral and neurological assessments at the Department of Pediatrics and the Department of Rehabilitation at Mie University Hospital.

### 2.5. Statistics

The comparison of Z-scores was according to the Japanese standard curve on the fetal biometric parameters at the start of tadalafil treatment (based on the time of diagnosis in the conventional treatment group) and at 2 weeks and 4 weeks of treatment and were performed within each group using the Wilcoxon signed-rank test (values at the start vs. 2 weeks after the start, at start vs. 4 weeks after the start, and at 2 weeks vs. 4 weeks after the start). The Wilcoxon signed-rank test was used to analyze the standard deviation of the biometrical parameters. The resistance index (RI) was analyzed from the Doppler assessment (UA-RI, MCA-RI) using the Wilcoxon signed-rank test. Analyses were performed using JMP^®^14.2.0 (SAS Institute Inc., Cary, NC, USA).

## 3. Results

### 3.1. Characteristics of Participants

Fifty pregnant women with FGR treated with tadalafil (tadalafil treatment group) and ten who received conventional management (control group) were included in the present study. The characteristics of the objectives are summarized in Table 1. The median maternal age at start of treatment was 33 years in the tadalafil group and 32 years in the control group. The median GA at start of treatment was 30 and 31 weeks in each group, and the median GA at delivery was 37 weeks in both groups. Twenty-four participants were primiparas, two smoked, and eight had HDP.

### 3.2. Fetal Biometric Measurements

The fetal biometric measurements at start of treatment are shown in Table 1. The median of EFBW at start of treatment was 1089 g (interquartile range (IQR): 748–1390) with a Z-score of −2.2 (IQR: −2.4 to −1.9) in the tadalafil group and 1348 g (IQR: 1148–1482) with a Z-score of −2.1 (IQR: −2.3 to −2.0) in the control group. The median Z-scores of BPD, HC, AC, and FL were −1.1, −0.8, −1.6, and −1.7, respectively, in the tadalafil group and −1.5, −0.4, −1.6, and −1.4, respectively, in the control group.

Figure 1 shows the results of Z-score of the fetal biometric measurements at the start of treatment and at 2 weeks and 4 weeks of treatment by box-and-whisker plot. Since patients who delivered after the start of treatment were excluded from the analysis due to lack of values, in the tadalafil treatment group, 43 patients were analyzed at 2 weeks of treatment, and 37 patients were analyzed at 4 weeks. In the control group, 10 patients were undelivered and eligible for analysis at 2 weeks of treatment, and 7 patients were analyzed at 4 weeks.

In the tadalafil treatment group, the Z-score of HC was significantly increased at 4 weeks of treatment (*p* = 0.005). There was no significant difference in the Z-scores between the start of treatment and 2 and 4 weeks of treatment in the BPD, FL, and EFBW, although AC tended to increase by 4 weeks after tadalafil treatment initiation (*p* = 0.06). HC/AC was significantly decreased at 4 weeks of tadalafil treatment. In the conventional treatment group, there were no significant differences in any biometric measurement.

The comparison of the Z-score of the biometric parameters at each dose of tadalafil (10 mg vs. 20 mg vs. 40 mg) is shown in the Appendix A. Post hoc analysis was not performed because there was no significant difference in the Kruskal–Wallis H test of the three-group analysis.

Appendix A shows the Z-score of the biometric parameters at each dose in the boxplots. In the tadalafil 20 mg group, a significant increase in HC was observed from the start of treatment to 4 weeks of treatment. A similar trend was also observed in the 40 mg group on HC, although the difference did not reach significance (*p* = 0.055).

### 3.3. Doppler Study

The results of the Doppler study are shown in Figure 2. The UA-RI significantly decreased after 4 weeks (*p* = 0.049) in the tadalafil group. In the control group, no significant changes in UA-PI were observed in the study period. MCA showed no significant change in either group. The result of the Doppler study at each dose are shown in the Appendix A. There were no significant changes both UA and MCA.

The neonatal characteristics of both groups are presented in the Appendix A. There were no significant differences between the two groups.

### 3.4. Maternal Hemodinamics

Maternal sBP, dBP, MAP, and HR as the evaluation of hemodynamics is shown in Table 2. There are significant decreases in dBP in the tadalafil treatment group at 2 weeks of treatment.

### 3.5. KSPD Assessment

The median DQ score was 89 for P–M, 91 for C–A, 94 for L–S, and 88 for total area at 1.5 years CA. At 3 years old, the respective scores were 96, 86, 92, and 87 (Table 3). The number of cases with a DQ ≤ 70 and therefore judged to be abnormal was seven (19%) for P–M, three (8%) for C–A, four (19%) for L–S, and four (11%) for total area at 1.5 years CA. At 3 years, old the respective scores were 3 (16%), 4 (21%), 3 (16%), and 3 (16%) (Table 4).

## 4. Discussion

This study is the first to evaluate the change in the biometrical measurement of tadalafil-treated fetuses with growth restriction. These findings are valuable because of the scarce available data on the effect of tadalafil on fetal biometric assessments and their contribution to elucidating tadalafil’s mechanisms in FGR.

In the case of FGR, the redistribution of blood flow to vital organs, including the brain myocardium, and adrenal gland occurs as a physiological response to insufficient fetal-placental perfusion. A representative finding is the “brain-sparing effect”, which indicates an increase in fetal cerebral blood flow [54]. We considered the effect of tadalafil on FGR by evaluating fetal growth, which might be affected by blood flow.

In this study, HC Z-score increased significantly 4 weeks after tadalafil administration, while no significant change was observed in the control group. Since it has been reported that delayed or arrested HC growth is one of the strongest factors influencing neurological development [41], the findings in this study are remarkable.

As already known, blood flow redistribution and growth stagnation appear in stages [41], and typically, AC growth is stagnant followed by stagnation of HC growth. From the results in this study, it is thought that tadalafil may have an effect in maintaining HC growth by improving blood flow. However, there is no evidence that maintenance of blood flow leads to the development of HC; therefore, this is speculative.

In a growth-restricted fetus, reduced umbilical vein flow volume leads to increased ductus venosus shunting from the liver to the heart [55]. Therefore, in general, the fetal liver size is decreased and lagging; consequently, AC growth is seen as a clinical sign reflecting decreased glycogen stores [56,57]. In this study, not only was the increase in HC observed, but HC/AC were significantly decreased, and the tendency for AC to increase was shown in the tadalafil group compared with the control group. This finding might suggest that the increased flow of the ductus venosus as “redistribution”, affected by the reduced umbilical vein, is mitigated, and blood flow to the liver is improved by tadalafil treatment.

The decrease in UA-RI in the tadalafil group, which reflects placental function, also supports this hypothesis. In FGR, UA-RI increases, reflecting high umbilical–placental vascular resistance [58,59]. The significant decrease in UA-RI that appears to follow the normal curve observed in this study might signify the maintenance of placental function. Increased umbilical vein blood volume due to improved utero–placental perfusion and mitigated blood flow redistribution with tadalafil treatment could have led to AC growth. These effects may contribute to the prolongation of gestational age in FGR, which was observed in TADAFER II [25].

The KSPD assessment is widely used in Japan for the evaluation of the development of high-risk infants; however, it is not used worldwide. The Bayley Scales of Infant Development (BSID) is a globally well-known and standard representative developmental assessment test. KSPD and BSID have been compared in two previous studies [60,61], which reported good correlation between the two tests. We previously reported on the favorable neurodevelopmental prognosis of tadalafil-treated children in a small number of cases [62].

The results of the KSPD test in this study were not able to indicate the efficacy of tadalafil because statistical analysis was not done to compare with the control group. However, the median score of each area in the tadalafil treatment group exceeded 85 (normal range), and the number of cases that had a DQ ≤ 70 were less than 20%, which was considered favorable. Few studies have examined KSPD scores for FGR in the past; however, Motoki et al. reported neurodevelopmental evaluation of severe FGR [63]. Severe FGR is defined as the estimated body weight in utero by ultrasonic measurement that is below the 3rd percentile (−1.88 SD), and the cases that delivered at less than 37 weeks were included. The scores of these cases were divided into three scoring groups (DQ ≥ 85, 70 < DQ < 85, and DQ ≤ 70), and the number of cases in each group was reported. When matching the subjects in our study with the criteria of Motoki et al. [58], the proportion of cases with DQ < 70 (abnormal) was similar (present study vs. data from Motoki et al.: P–M: 18.2% (2/11) vs. 26.9% (7/26); C–A: 9.1% (1/11) vs. 12.0% (3/25); L–S: 18.2% (2/11) vs. 31.7% (8/26); total area; 9.1% (1/11) vs. 15.4% (4/26)). Although it cannot be concluded that tadalafil has a positive effect, at least the outcome was not poor. Evaluation throughout school age, adolescence, and adulthood is necessary, and we hope to continue to follow-up the children. Evaluation throughout school age, adolescence, and adulthood is necessary, and we hope to continue the follow up not only for neurodevelopmental prognosis but also for adult diseases such as diabetes, cardiovascular disorders, and hyperlipidemia, which are well known to be associated with FGR [64].

In the present study, the number of cases in the control group was limited to 10. This study was conducted at a single center, and in the same period, the number of cases without tadalafil treatment was limited. For a more robust future study, the number of control cases would need to be similar to that of the experimental group. Nonetheless, this contrasting result between the two groups is a reason for the prolongation of pregnancy in FGR due to tadalafil treatment and is a matter of great interest.

Although FGR is said to be affected by social background including economic, diet, and regional factors [65], no data were available for social background in this study, including the sociodemographic, psychosocial, psychological, clinical, and dietary variables. Maternal BMI, smoking status, and HDP that might affect the fetal growth are summarized in the maternal characteristics (Table 1). There are no significant differences in the points. In the future, it is necessary to investigate the association between the patient’s background and the efficacy of tadalafil treatment for FGR.

The total number of cases with doses of tadalafil 10 mg, 20 mg, and 40 mg were analyzed in this study, and the dose dependence was problematic. As shown in Figure 2, HC Z-scores tended to increase in both the 20 mg and 40 mg groups. Therefore, currently, we considered that there is no dose dependency.

## 5. Conclusions

In this study, tadalafil significantly increased the HC of growth-restricted fetuses. This effect may result from improved uteroplacental perfusion and the maintained cerebral blood flow redistributed as a brain-sparing effect. These findings may contribute to the prolongation of gestation of fetuses with growth restriction.

## Figures and Tables

**Figure 1 medicina-59-00900-f001:**
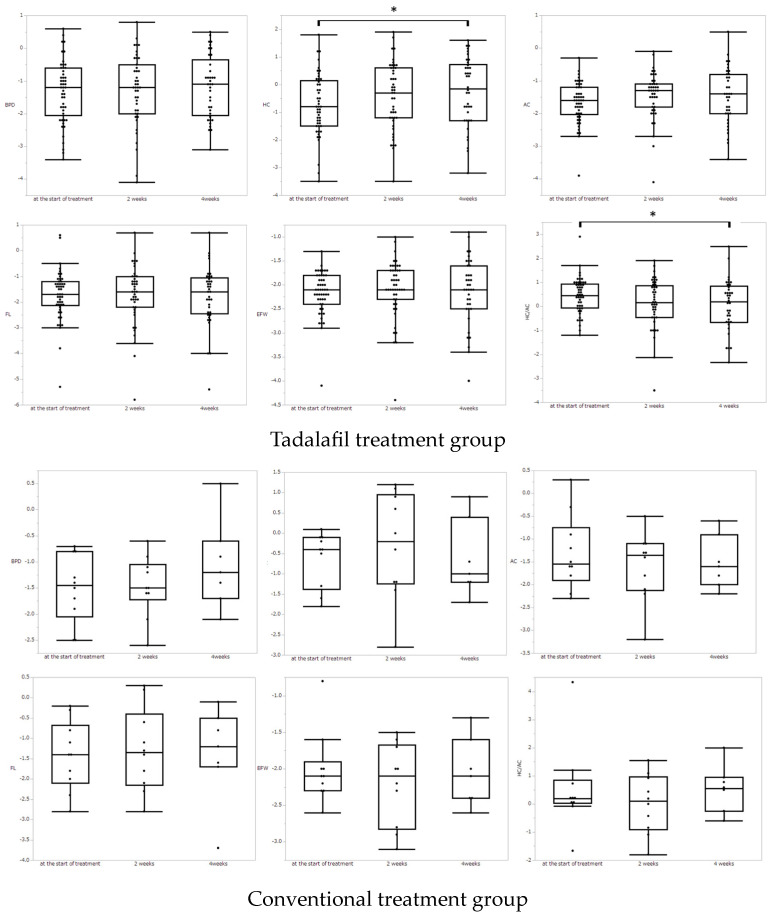
The comparison of Z-score for each biometric parameter at the start of treatment vs. 2 weeks after the start, at start vs. 4 weeks after the start, and at 2 weeks vs. 4 weeks after the start. Asterisks show statistically significant differences (*p* < 0.05) by a paired test using the Wilcoxon signed-rank test.

**Figure 2 medicina-59-00900-f002:**
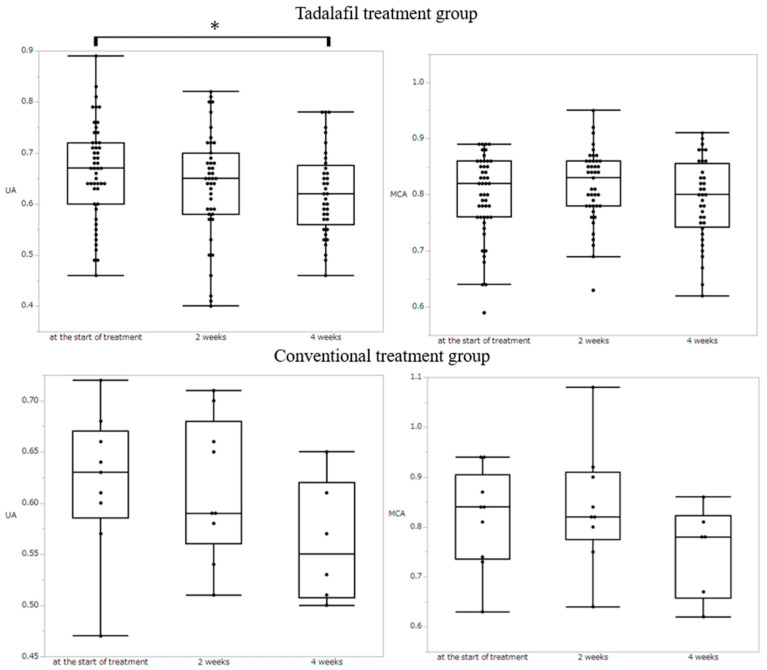
Doppler assessment of UA-RI and MCA-RI (at the start of treatment vs. 2 weeks after the start, at start vs. 4 weeks after the start, and at 2 weeks vs. 4 weeks after the start.). Asterisks show statistically significant differences (*p* < 0.05) by a paired test using the Wilcoxon signed-rank test.

**Table 1 medicina-59-00900-t001:** Patient and Fetal Characteristics.

	Tadalafil Group (*n* = 50)	Control Group (*n* = 10)
Maternal characteristics		
Age (y)	33 (28–35)	32 (30–36)
Height (cm)	156 (151–160)	156 (154–162)
BMI	21.1 (19.8–23.1)	19.8 (21.0–18.5)
Body weight before pregnancy (kg)	50.0 (46.0–55.6)	48.5 (46.3–51.7)
Primipara	24	2
Smoking	2	1
Gestational age at the start of treatment (weeks)	30 (27–32)	31 (30–32)
Incidence of HDP	8	1
Gestational age at delivery (weeks)	37 (35–37)	37 (34–38)
Fetal characteristics		
EFBW at the start of treatment (g)	1089 (748–1389)	1348 (1148–1482)
Z-score	−2.2 (−2.4 to −1.9)	−2.1 (−2.3 to −2.0)
BPD at the start of treatment (cm)	7.03 (6.45–7.75)	7.12 (6.97–7.55)
Z-score	−1.1 (−1.9 to −0.63)	−1.5 (−1.9 to −0.9)
HC at the start of treatment (cm)	25.8 (22.8–27.1)	26.5 (26.2–27.6)
Z-score	−0.8 (−1.5–0.1)	−0.4 (−1.1 to −0.1)
AC at the start of treatment (cm)	21.7 (19.1–24.0)	23.8 (22.3–25.3)
Z-score	−1.6 (−2 to −1.2)	−1.6 (−1.8 to −1.0)
FL at the start of treatment (cm)	4.93 (4.27–5.32)	5.34 (5.00–5.53)
Z-score	−1.7 (−2.1 to −1.3)	−1.4 (−2.0 to −0.9)

Data are reported as median (interquartile range) BPD, biparietal diameter; AC, abdominal circumference; HC, head circumference; FL, femur length; EFBW, estimated fetal weight.

**Table 2 medicina-59-00900-t002:** Maternal Hemodynamics.

Maternal Vital Sign		Tadalafil Group (*n* = 50)	Conventional Treatment Group (*n* = 10)	*p*-Value
At the start of treatment	sBP (mmHg)	108 (102–120)	108.5 (103.75–121.75)	0.76
	dBP (mmHg)	62 (56–69)	71 (59.25–78)	0.86
	MAP (mmHg)	77.33 (71.33–85)	85 (72.75–93)	0.18
	HR (bpm)	78 (70–90)	83.5 (76.25–88.5)	0.32
2 weeks of treatment	sBP (mmHg)	106.5 (100–112.25)	110 (98.75–119.5)	0.64
	dBP (mmHg)	58.5 (50.75–65)	69 (59.75–74.25)	0.018
	MAP (mmHg)	75.33 (67.83–79.75)	84 (72.5–88.5)	0.076
	HR (bpm)	76 (70–84.25)	74 (68.5–78.5)	0.56
4 weeks of treatment	sBP (mmHg)	107 (102.5–114.5)	103 (99–113)	0.30
	dBP (mmHg)	59.5 (55.25–66.75)	60 (56–66)	0.87
	MAP (mmHg)	75 (70.83–83.67)	73 (72–82)	0.62
	HR (bpm)	76 (67.25–81.75)	70 (66–82)	0.47

Data are shown as median (IQR). Analyses were done by Wilcoxon rank-sum test.

**Table 3 medicina-59-00900-t003:** DQ score by KSPD at the 1.5 years of CA and 3 years old.

	1.5 Years of CA	3 Years Old
	(*n* = 37)	(*n* = 19)
P–M	89 (76–94)	96 (77–102)
C–A	91 (80–102)	86 (79–96)
L–S	94 (84–99)	92 (77–96)
Total	88 (82–100)	87 (76–98)

Data are shown as median DQ (IQR). KSPD, Kyoto Scale of Psychological Development; P–M, postural–motor; C–A, cognitive–adaptive; L–S, language–social; CA, collected age.

**Table 4 medicina-59-00900-t004:** Number and percentage of cases in each scoring area.

**1.5 Years CA (*n* = 37)**				
**DQ**	**P–M**	**C–A**	**L–S**	**Total Area**
<70	7 (19%)	3 (8%)	7 (19%)	4 (11%)
70–85	10 (27%)	10 (27%)	4 (11%)	10 (27%)
>85	20 (54%)	24 (65%)	26 (70%)	23 (62%)
		Data are presented as *n* (%)		
**3 Years Old (*n* = 19)**				
**DQ**	**P–M**	**C–A**	**L–S**	**Total Area**
<70	3 (16%)	4 (21%)	3 (16%)	3 (16%)
70–85	4 (21%)	2 (11%)	5 (26%)	5 (26%)
>85	12 (63%)	13 (68%)	11 (58%)	11 (58%)
		Data are presented as *n* (%)		

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
