# Peer review of "Fetal Biometric Assessment and Infant Developmental Prognosis of the Tadalafil Treatment for Fetal Growth Restriction"

_medicina, 2023, doi:10.3390/medicina59050900_

Round 1
Reviewer 1 Report
1.Introduction The benefit of sildenafil has not been demonstrated in studies. What is the difference between sildenafil and tadalafil in terms of mechanism of action. Based on which hypothesis, it was concluded that it could be beneficial on FGR in ıntroduction section.
2.Is stadalafil routinely started in pregnant women with FGR? Otherwise, the conditions under which treatment is started, why the study was designed retrospectively, and its prospective case-control would make the study more valuable.
3.The number of patients in the control group is very small, which reduces the reliability of the study.
4.Units with maternal characteristics should be added in Table 1
5. The number of patients in the control group is very small, which reduces the reliability of the study. However, it contributes to science in terms of being one of the rare studies on this subject and following the neurological development of fgr fetuses at 1.5 and 3 years of age.
Author Response
・Introduction The benefit of sildenafil has not been demonstrated in studies. What is the difference between sildenafil and tadalafil in terms of mechanism of action. Based on which hypothesis, it was concluded that it could be beneficial on FGR in ıntroduction section.
Thank you for your insightful comment. Tadalafil and sildenafil differ in terms of pharmacological function such as half-life, selectivity for PDE5, and placental transmission, and we consider that they have different effects on the fetus.
We have added the following text in the Introduction section;
“Compared with sildenafil, tadalafil have longer half-life and a more rapid onset of action [31], and less susceptibility to the intake of a high-fat meal [32,33]. In addition, it is thought that tadalafil and sildenafil have different function for feto-placental perfusion. Walton et al reported that sildenafil citrate reversed preconstricted placental–fetal arterial perfusion in an ex-vivo human placental model, whereas tadalafil produced no response [34]. In our mice experiment, width of the maternal blood sinuses in placenta was significantly dilated by tadalafil administration; in contrast there was no change in fetal capillary width with the administration [28]. This indicates that tadalafil does not cross the human placental barrier, or is degraded by trophoblast cells, and that the difference in the effects between tadalafil and sildenafil in the fetus via the placenta might result in differences in safety and efficacy.”
・Is tadalafil routinely started in pregnant women with FGR? Otherwise, the conditions under which treatment is started, why the study was designed retrospectively, and its prospective case-control would make the study more valuable.
Thank you for your comment.
The subjects of this study were those who had already been enrolled in other studies, or those who were using tadalafil at their request. The cases in this study were those who had already been enrolled in another study, Therefore, it has a retrospective study design. As you said, a prospective study is needed to establish further evidence. A placebo-controlled, randomized phase II study is currently ongoing.
We have added the following sentences in the Materials and Methods section;
“This study included cases registered in our previous retrospective study [26] and in the phase I study [27]. Additional cases in which tadalafil had been administered upon the request of patients and had not been included in the two previous studies were also included in the present study.”
・The number of patients in the control group is very small, which reduces the reliability of the study.
As you pointed, this is one of the limitations of this study. Because of the limited number of patients without tadalafil treatment for FGR in our hospital during the study period, the number of eligible cases was limited to 10, as described in the DISCUSSION section.
・Units with maternal characteristics should be added in Table 1
Thank you for your advice.
The units have been added to the maternal characteristics in Table 1.

Reviewer 2 Report
The authors have done retrospective study on 50 fetuses and 10 controls between 2015 to 2019 to evaluate the fetal biometric assessment and infant developmental prognosis of the tadalafil treatment for fetal growth restriction to see its effects on the still birth and neonatal morbidity with favourable results.The introduction is very detailed, provide sufficient background and include all relevant references. The research design appropriate for a retrospective study and can make it a baseline study and carry on with prosepective study with more rigor and perhaps randomisation to make it more evidence based. The materials and methods adequately described with details. The results clearly presented and statistical analysis areappropriately used. The conclusion is very well supported bythe results achieved. The references are all recent and relevant and in the journal style. The tables and illustrations are appropriate and supplementing the written text. Overall the article is good. However, the english language, grammar, punctuation, etc can be improved by profeesional editing service or by person whose English language is a first language.
Author Response
・The English language, grammar, punctuation, etc can be improved by profeesional editing service or by person whose English language is a first language.
Thank you for your advice. We have re-submitted our revised manuscript to an English editing service provider.

Reviewer 3 Report
The manuscript entitled “Fetal biometric assessment and infant developmental prognosis of the tadalafil treatment for fetal growth restriction” by Tsuji et al assesses the effect of Tadalafil on fetal growth under the justification that fetal growth restriction leads to outcomes that are negative for the child`s health later on postnatal life. After reading carefully the manuscript, I found its epistemic structure poorly substantiated and inadequately fabricated. They are not “correcting” fetal growth restriction but placental insufficiency, and the presumed therapeutic benefits of tadalafil on fetal growth are rather indirect after assuming improved circulation in placental vessels following vasodilation. In addition, this work lacks scholarly. The introduction is formed by a narration of facts divided in three sections. None of them presents the conceptual model that is used as a guide to design the study. No premises or predictions are made and tested. This is important since the model justifies scientifically the therapeutic measure advised or used. I recommend the authors to build up their manuscript around the model that explains how placental insufficiency occurs, and use fetal growth as an index of placental functional improvement. This is even more important since there are numerous previous studies that contradict the central tenet of this work, none of which are critically commented in the discussion under the light of the reported findings. In addition, to discuss such contradictions under the light of the etiological model of placental insufficiency could be particularly instructive.
With regard to methods, I strongly encourage authors to include and analyze (multiple correlation analyses), in the context of their work, the sociodemographic, psychosocial, psychological, clinical and dietary variables of all mothers considered in the study as gestational stress plays a central role in producing placental insufficiency and fetal growth retardation. In addition, please explain why authors used three different doses of tadalafil and how such doses were estimated. They also should disclose the posology and route and hour of administration. Please, explain why normal pregnancies, as a true reference, were excluded from the study. I also think that authors must make available, as a supplementary material, the letter issued by the ethical committee authorizing the realization of this project. Also, the informed consent forms of each participant must be available.
On the other hand, I am not convinced that the use of the Wilcoxon signed-rank test is adequate to conduct the statistical analyses since sample size is different in both groups. In general, the Wilcoxon signed-rank test compares whether differences between pairs of data follow a symmetric distribution around a value. This condition is not surpassed by data provided in this study (tadalafil treated n=50 or less; conventional treatment n=10 or less); values are not pair-wise. Authors must disclose the reason why some participants drop off the study. In addition, authors mentioned they used three different doses of tadalafil. Yet, they seem to have all tadalafil treated women grouped and considered as equal. No data on patients treated with 10 mg of tadalifil were introduced in the manuscript. They should have compared 10, 20 and 40mg tadalafil treated women as independent groups, and then with women that underwent conventional management. Also, authors must report and consider the mother biometric data at 2 and 4 weeks of treatment and to evaluate the impact of the mother’s condition on the observed results; obviously tadalafil is not affecting the hemodynamics of the placenta only. And, speaking on these matters, it is also intriguing that authors did not report ultrasound data (e.g., Doppler studies) of the placentas to demonstrate the alluded effect. Finally, it is well documented that interventions during gestation condition the state of health and disease in adulthood (i.e., developmental origin of adult diseased states). Hence, assessing potentially harmful effects of tadalafil when the kids (by the way no sex is reported and no sex differences were evaluated) have three years of age is largely insufficient, and meaningless if authors presumed to have ruled out long-term collateral damaging effects of tadalafil from this age onwards by using physical, intellectual, emotional tests. In sum, the experimental design seems inadequate.
Regarding the results, I believe section 3.2 must be revised and re-edited. English must be corrected. Specifically, page five, lines 179-184 are confusing. Notice, for instance, the first sentence of the following paragraph; seems contradictory.
“HC significantly increased 4 weeks after the start of treatment (p = 0.005), in contrast with the control group, but with no significant difference. There was no significant difference between the
start of treatment and after 2 and 4 weeks on the Z-scores of BPD, FL, and EFBW in both groups, although AC tended to increase 4 weeks after tadalafil treatment initiation (p = 0.06). HC/AC was significantly decreased at 4 weeks from the start of tadalafil treatment, while there was no significant difference in the control group”
I recommend authors to conduct principal component analyses and hierarchical clustering dendrograms to qualitatively assess the data. In addition, instead of deploying the data in tables, please, from table 2 to 5, transform each in whisker and box plots. Each plot must “dot” all of the individuals included in the analyses through the corresponding boxes.
Finally, I believe that the discussion should be re-conceptualized and re-written based on the recommendations made. It still is speculative especially with regard to the differential effect observed throughout the fetal body. The alluded “mechanism” intended to explain why the cranium/brain response is much robust as compared to other body parts and organs are at best an unsubstantiated hypothesis. Authors should also dedicate some words to explain why the response reported had no dose dependent effects. All this justifies the inclusion of the model of the diseased state since it will help in providing support, order, cohesion and coherence to the discussion and the paper as a whole.
Minor concerns:
English editing needed. If authors paid for English editing, there is no need to thank the company responsible for doing it. Authors must check that the editing service is doing its job well. There are numerous mistakes that cannot be tolerated if the service is offering a high quality work.
Mandatory
Ethical issues that must be addressed:
It is intriguing that the study is said to have been conducted between July 2015 and September 2019, but the letter that authorized the study to be pursued was issued by the ethical committee in March 2021. If this is true, the letter was obtained way after the study has already started and concluded; a highly irregular situation that must be revised and cleared out. Please, in the section corresponding to the informed consent statement, disclose (make explicit) the guidelines and regulations that were relevant for the study.
Author Response
・They are not “correcting” fetal growth restriction but placental insufficiency, and the presumed therapeutic benefits of tadalafil on fetal growth are rather indirect after assuming improved circulation in placental vessels following vasodilation. In addition, this work lacks scholarly. The introduction is formed by a narration of facts divided in three sections. None of them presents the conceptual model that is used as a guide to design the study.
No premises or predictions are made and tested. This is important since the model justifies scientifically the therapeutic measure advised or used. I recommend the authors to build up their manuscript around the model that explains how placental insufficiency occurs, and use fetal growth as an index of placental functional improvement. This is even more important since there are numerous previous studies that contradict the central tenet of this work, none of which are critically commented in the discussion under the light of the reported findings. In addition, to discuss such contradictions under the light of the etiological model of placental insufficiency could be particularly instructive.
Thank you for your insightful comment.
Accordingly, we hypothesized that tadalafil works with an indirect effect by improving placental circulation rather than a direct effect on the fetuses. In previous studies we verified its efficacy and safety (25-30). However, the description in the introduction is insufficient, so we have re-written why fetal assessment is an indicator of placental dysfunction, taking into account the pathogenesis of placental dysfunction. Despite the negative results with sildenafil in previous studies, we also add a note on the differences between sildenafil and tadalafil and explained the safety of this study.
We have added the following text in the introduction section;
“We hypothesized that tadalafil improves fetoplacental perfusion on FGR. In placental insufficiency, there is histological evidence of impaired trophoblastic invasion to spiral arteries in the placental bed which lead to narrowed spiral arteries [35]. Fetal hypoxemia produces a physiological response that increases blood flow to vital organs, including the brain, myocardium, and adrenal gland, as a “redistribution” [36–40]. The mechanism causes lowered abdominal circumference (AC) of fetuses due to reduced blood flow to the liver and glycogen stores. As gestational age progress and the mechanism cannot kept up, it leads to stagnation of the head circumference (HC) growth and this is thought to be associated with a worse neurological prognosis, including psychomotor and cognitive retardation in the child [41]. Therefore, if tadalafil causes an improvement in uteroplacental perfusion, changes in hemodynamics may affect the fetal growth pattern including AC and HC.”
As you mentioned, many previous studies, i.e., on sildenafil showed negative data. We consider tadalafil and sildenafil have diffrent function for feto-placental perfusion. Walton et al reported that sildenafil citrate reversed preconstricted placental–fetal arterial perfusion in an ex-vivo human placental model, whereas tadalafil produced no response. In our mice experiment, width of the maternal blood sinuses in placenta was significantly dilated by tadalafil administration; in contrast there was no change in fetal capillary width with the administration. This indicates that tadalafil does not cross the human placental barrier, or is degraded by trophoblast cells, and that the difference in the effects between tadalafil and sildenafil in the fetus via the placenta might result in differences in safety and efficacy.
We also added the following text in the introduction section
“Compared with sildenafil, tadalafil have longer half-life and a more rapid onset of action [31], less susceptible to the intake of a high-fat meal [32,33]. In addition, it is thought that tadalafil and sildenafil have different mechanism for feto-placental perfusion. Walton et al reported that sildenafil citrate reversed preconstricted placental–fetal arterial perfusion in an ex-vivo human placental model, whereas tadalafil produced no response [34]. In our mice experiment, width of the maternal blood sinuses in placenta was significantly dilated by tadalafil administration; in contrast there was no change in fetal capillary width with the administration [28]. This indicates that tadalafil does not cross the human placental barrier, or is degraded by trophoblast cells, and that the difference in the effects between tadalafil and sildenafil in the fetus via the placenta might result in differences in safety and efficacy.
For that reason, research for the efficacy of tadalafil on FGR has continued as agents other than sildenafil yielded negative results.”
We have added the following references:
- Rotella, D.P. Phosphodiesterase 5 inhibitors: current status and potential applications. Nat Rev Drug Discov. 2002,1,674–82.
- Wilkins, M.R, Wharton, J, Grimminger, F, et al. Phosphodiesterase inhibitors for the treatment of pulmonary hypertension. Eur. Respir. J. 2008,32,198–209.
- Forgue, S.T, Patterson, B.E, Bedding, A.W, et al. Tadalafil pharmacokinetics in healthy subjects. Br. J. Clin. Pharmacol. 2006,61,280–288.
- Walton, R.B, Reed, L.C, Estrada S.M, et al. Evaluation of sildenafil and tadalafil for reversing constriction of fetal arteries in a human placenta perfusion model. Hypertension, 2018, 72,167–176.
35. Khong, T.Y, De Wolf, F, Robertson, W.B, et al. Inadequate maternal vascular response to placentation in pregnancies complicated by pre-eclampsia and by small-forgestational age infants. Br. J. Obstet. Gynaecol. 1986,93,1049–1059.
・With regard to methods, I strongly encourage authors to include and analyze (multiple correlation analyses), in the context of their work, the sociodemographic, psychosocial, psychological, clinical and dietary variables of all mothers considered in the study as gestational stress plays a central role in producing placental insufficiency and fetal growth retardation.
Accordingly, it has been reported that economic and social reasons increase the risk of developing FGR. Unfortunately, because we could not collect such data in this study, they were not evaluated.
This study was conducted at a single center, with comparable management and no regional differences among participants. Maternal BMI, smoking status, and hypertensive disorders of pregnancy which are considered to affect the fetal growth are summarized in maternal background. There were no significant differences in these items between the two groups. Based on your comment, we have added limitation on the social background of the participants.
We have added the following sentences in the Discussion section:
“Although FGR is said to be affected by social background including economic, diet and regional factors [65], no data was available for social background in this study including the sociodemographic, psychosocial, psychological, clinical, and dietary variables. Maternal BMI, smoking status, and hypertensive disorders of pregnancy that might affect the fetal growth are summarized in maternal characteristics (Table 1). There are no significant differences in the point. In the future, it is needed to investigate the association between the patient's background and the efficacy of tadalafil treatment for FGR.”
We have added the following references:
- Karmer MS. The epidemiology of low birthweight. Nestle Nutr Inst Workshop Ser. 2013;74:1-10.
・In addition, please explain why authors used three different doses of tadalafil and how such doses were estimated. They also should disclose the posology and route and hour of administration.
Thank you for your suggestion.
The clinical study of tadalafil was originally inspired by a report [42] showing favorable fetal development in pregnancies with pulmonary hypertension with a high incidence of FGR. Insurance coverage in Japan for pulmonary hypertension covered tadalafil doses of 20 mg and 40 mg, which are known to be safe for pregnant women, and 10 mg which is indicated for erectile dysfunction. Therefore, 10 mg, 20 mg, and 40 mg of tadalafil have been used in studies. Tadalafil was orally administered at 11:00 every day until delivery.
We have added the following sentence in the Materials and Methods section:
“The clinical study of tadalafil was originally inspired by a report [42] showing favorable fetal development in pregnancies with pulmonary hypertension with a high incidence of FGR. Insurance coverage in Japan for pulmonary hypertension covered for tadalafil doses of 20 mg and 40 mg which are known to be safe for pregnant women, and 10 mg indicated for erectile dysfunction. Therefore, 10 mg, 20 mg and 40 mg of tadalafil have been used in studies. Tadalafil was orally administered at 11:00 a.m. every day until delivery.”
We have added following references:
- Daimon A, Kamiya CA, Iwanaga N, et al. Management of pulmonary vasodilator therapy in three pregnancies with pulmonary arterial hypertension. J Obstet Gynaecol Res. 2017 May;43(5):935-938.
・Please, explain why normal pregnancies, as a true reference, were excluded from the study. I also think that authors must make available, as a supplementary material, the letter issued by the ethical committee authorizing the realization of this project. Also, the informed consent forms of each participant must be available.
Thank you for your comment.
Since this is a biometric assessment of FGR cases treated with tadalafil, we considered it appropriate to compare FGR cases as controls, and excluded normal pregnancies.
We submit the research protocol converted to PDF. Only Japanese version is available.
Informed consent was obtained in the form of opt-out on the web-site, as described in Informed Consent Statement section. We also submit the form of opt-out.
・On the other hand, I am not convinced that the use of the Wilcoxon signed-rank test is adequate to conduct the statistical analyses since sample size is different in both groups. In general, the Wilcoxon signed-rank test compares whether differences between pairs of data follow a symmetric distribution around a value. This condition is not surpassed by data provided in this study (tadalafil treated n=50 or less; conventional treatment n=10 or less); values are not pair-wise. Authors must disclose the reason why some participants drop off the study.
Thank you for your insightful comment.
The analysis in Tables 2 and 3 compares the Z-scores of the fetal biometric parameters within same group at the start of tadalafil treatment (based on the time of diagnosis in the conventional treatment group), 2 weeks of treatment, and 4 weeks of treatment, using the Wilcoxon signed-rank test as paired test. The comparison between the two groups were not performed.
Therefore, the data after 2 weeks and 4 weeks show the number of cases in which pregnancy continued until that point, not drop-out cases.
We apologize for the confusing description in the Statistics section. The contents have been changed as follows.
“The comparison of Z-scores was according to the Japanese standard curve on the fetal biometric parameters at the start of tadalafil treatment (based on the time of diagnosis in the conventional treatment group), and at 2 weeks and 4 weeks of treatment; and were performed within each group using the Wilcoxon signed-rank test (values at the start vs. 2 weeks after the start, at start vs. 4 weeks after the start, and at 2 weeks vs. 4 weeks after the start).”
・In addition, authors mentioned they used three different doses of tadalafil. Yet, they seem to have all tadalafil treated women grouped and considered as equal. No data on patients treated with 10 mg of tadalafil were introduced in the manuscript. They should have compared 10, 20 and 40mg tadalafil treated women as independent groups, and then with women that underwent conventional management.
Thank you for your insightful comment.
Since there were only 3 cases of tadalafil 10 mg administration, it was not included in the analysis by dose shown in table S1. We have changed table S1 to include tadalafil 10 mg group as three-group analysis. Post hoc analysis was not performed because there was no significant difference in the Kruskal-Wallis H test.
Since we mainly performed the comparison within each group (start of the treatment vs. 2, 4 weeks after the start) by a paired test using the Wilcoxon signed-rank test. Analyzes for each dose group has been added to the Figure S1 as boxplots. There were only 3 cases in which 10 mg tadalafil was administered, and it is considered not worthy of analysis, so the boxplot was also omitted.
We have added the following word and sentences in the Results section:
“The comparison of the Z-score of each biometric parameter for each dose of tadalafil (10 mg vs. 20 mg vs. 40 mg) are shown in the supplemental appendix (Table S1). Post hoc analysis was not performed because there was no significant difference in the Kruskal-Wallis H test of the three-group analysis.
Figure S1 shows the Z-score of the biometric parameters at each dose in the boxplots. In the tadalafil 20 mg group, a significant increase in HC was observed from the start of treatment to 4 weeks of treatment. A similar trend was also observed in the 40 mg group on HC, although the difference did not reach significance (p=0.055).”
・Also, authors must report and consider the mother biometric data at 2 and 4 weeks of treatment and to evaluate the impact of the mother’s condition on the observed results; obviously tadalafil is not affecting the hemodynamics of the placenta only. And, speaking on these matters, it is also intriguing that authors did not report ultrasound data (e.g., Doppler studies) of the placentas to demonstrate the alluded effect.
Thank you for your insightful comment.
We analyzed the maternal systolic blood pressure (sBP), diastolic blood pressure (dBP), mean arterial pressure (MAP), and heart rate (HR) at the start of treatment, and at 2 weeks, and 4 weeks of treatment to examine the effects of tadalafil on the maternal hemodynamics. We have added a table in the Result section (Table 2). There were no significant differences in these contents.
We also added the following sentences in the Materials:
“Maternal Hemodynamics
Maternal systolic blood pressure (BP), diastolic BP, mean arterial blood pressure (MAP), and heart rate (HR) before the start of treatment, and at 2 weeks, and 4 weeks of treatment were assessed to evaluate the change in maternal hemodynamics due to maternal tadalafil administration. All measurements were conducted with participants in the sitting position after at least 5 minutes of rest using automated, non-invasive blood pressure monitoring devices (HBP-1600®, OMRON, Japan). Data after waking up and before taking oral medication were collected.”
And also added the following sentences in the Results section:
“Maternal sBP, dBP, MAP, and HR as the evaluation of the hemodynamics is shown in table 2. There is significant decrease in dBP in the tadalafil treatment group at 2 weeks of treatment.”
・Finally, it is well documented that interventions during gestation condition the state of health and disease in adulthood (i.e., developmental origin of adult diseased states). Hence, assessing potentially harmful effects of tadalafil when the kids (by the way no sex is reported and no sex differences were evaluated) have three years of age is largely insufficient, and meaningless if authors presumed to have ruled out long-term collateral damaging effects of tadalafil from this age onwards by using physical, intellectual, emotional tests. In sum, the experimental design seems inadequate.
Thank you for your insightful comment. Accordingly, it has been proposed that in FGR, the environment in utero influence the onset of later adult diseases [64]. However, many studies report impaired neurodevelopment of infants in childhood [41,63]. We consider that it is important to evaluate the neurodevelopmental prognosis of tadalafil-treated child at early stage of 1.5 years and 3 years because there are evidence that it is poor in cases of FGR. We hope to assess not only at 1.5 and 3 years of age, but also more long-term prognosis including school age, adolescence and adulthood in the future.
We have added the following text in the Discussion section:
“Evaluation throughout school age, adolescence, and adulthood is necessary and we hope to continue to follow up the children. Evaluation throughout school age, adolescence, and adulthood is necessary, and we hope to continue the follow up not only for neurodevelopmental prognosis but also for adult diseases such as diabetes, cardiovascular disorders, and hyperlipidemia which are well known to be associated with FGR [64].”
We have added a reference:
- Baker, D.J, Winter, P.D, Osmond, C, et al. Weight in infancy and death from ischaemic heart disease. Lancet. 1989 Sep 9;2(8663):577-80.
・Regarding the results, I believe section 3.2 must be revised and re-edited. English must be corrected.
We apologize for the confusing text. Please note that it is not a comparison between groups, but a comparison within each group by a paired test using the Wilcoxon signed-rank test as described above. The followings have been corrected in the text:
“Figure 1 shows the results of Z-score of the fetal biometric measurements at the start of treatment, and at 2 weeks, and 4 weeks of treatment by box-and-whisker plot. Since patients who delivered after the start of treatment were excluded from the analysis due to lack of values, in the tadalafil treatment group, 43 patients were analyzed at 2 weeks of treatment, and 37 patients were analyzed at 4 weeks. In the control group, 10 patients were undelivered and eligible for analysis at 2 weeks of treatment, and 7 patients were analyzed at 4 weeks.
In the tadalafil treatment group, the Z-score of HC was significantly increased at 4 weeks of treatment (p=0.005). There was no significant difference in the Z-scores between the start of treatment and 2- and 4-weeks of treatment in the BPD, FL, and EFBW; although AC tended to increase by 4 weeks after tadalafil treatment initiation (p=0.06). HC/AC was significantly decreased at 4 weeks of tadalafil treatment. In conventional treatment group, there were no significant differences on all biometric part.”
・Instead of deploying the data in tables, please, from table 2 to 5, transform each in whisker and box plots. Each plot must “dot” all of the individuals included in the analyses through the corresponding boxes.
Thank you for your valuable feedback.
Tables 4 and 5, which show the results of KSPD, are data only for the tadalafil treatment group, there is no data for the conventional treatment group, and no comparison between groups was performed. Therefore, we have transformed Table 2 and Table 3 to whisker and box plots (Figure 1, 2).
・Finally, I believe that the discussion should be re-conceptualized and re-written based on the recommendations made. It still is speculative especially with regard to the differential effect observed throughout the fetal body. The alluded “mechanism” intended to explain why the cranium/brain response is much robust as compared to other body parts and organs are at best an unsubstantiated hypothesis. Authors should also dedicate some words to explain why the response reported had no dose dependent effects. All this justifies the inclusion of the model of the diseased state since it will help in providing support, order, cohesion and coherence to the discussion and the paper as a whole.
Your point is valid, and we agree that the function of tadalafil mentioned in the manuscript is only speculative. In the Introduction section, we have added the description about the pathogenesis of placental insufficiency, fetal hypoxia due to fetal placental ischemia, and "redistribution" of blood flow to fetal vital organs, and why we come up with the research proposal which evaluated fetal biometric parameters.
In the discussion section, the argument that the prolongation of gestational age reported in the TADAFER II study was associated with an increase in HC was highly speculative; therefore the sentence has been deleted. Also, I thought that the degree of speculation was too strong, so we have changed the Discussion text as follows:
“As already known, blood flow redistribution and growth stagnation appear in stages [41] and typically AC growth is stagnant followed by stagnation of HC growth. From the results in this study, it is thought that tadalafil may have an effect in maintaining HC growth by improving blood flow.”
In terms of the dose dependance, as mentioned above, we analyzed the group of tadalafil 20 mg and 40 mg, and found a trend toward an increase in HC Z-score, suggesting that both were effective in increasing HC. Therefore, currently, we considered that there is no dose dependence.
The following text was added to the Discussion section:
“The total number of cases with doses of tadalafil 10 mg, 20 mg, and 40 mg were analyzed in this study, and the dose-dependence was problematic. As shown in Figure 2, HC Z-scores tended to increase in both the 20 mg and 40 mg groups. Therefore, at present, it is considered that there is no dose-dependency.”
・English editing needed. If authors paid for English editing, there is no need to thank the company responsible for doing it. Authors must check that the editing service is doing its job well. There are numerous mistakes that cannot be tolerated if the service is offering a high quality work.
Thank you for your advice. We will ask the English proofreading service to re-proof.
・It is intriguing that the study is said to have been conducted between July 2015 and September 2019, but the letter that authorized the study to be pursued was issued by the ethical committee in March 2021. If this is true, the letter was obtained way after the study has already started and concluded; a highly irregular situation that must be revised and cleared out. Please, in the section corresponding to the informed consent statement, disclose (make explicit) the guidelines and regulations that were relevant for the study.
This study included cases registered in our previous retrospective study and in the phase I study. Additional cases in which tadalafil had been administered upon the request of patients and had not been included in the two previous studies were also included in the present study. This is a retrospective study that collected information only from medical records, and as stated in the informed consent statement, consent was obtained by optout on the website.
The guidelines and regulations that are relevant for the study is posted on the website of the Ministry of Health, Labor and Welfare (https://www.mhlw.go.jp/content/001077424.pdf). Unfortunately, only the Japanese version is available. We have added the URL of the guideline on Informed Consent Statement section.
We have added the following sentences in the Materials and Methods section:
“This study included cases registered in our previous retrospective study [26] and in the phase I study [27]. Additional cases in which tadalafil had been administered upon the request of patients and had not been included in the two previous studies were also included in the present study.”

Round 2
Reviewer 3 Report
No further changes are required